# The Inter-Organ Crosstalk Reveals an Inevitable Link between MAFLD and Extrahepatic Diseases

**DOI:** 10.3390/nu15051123

**Published:** 2023-02-23

**Authors:** Tsubasa Tsutsumi, Dan Nakano, Ryuki Hashida, Tomoya Sano, Machiko Kawaguchi, Keisuke Amano, Takumi Kawaguchi

**Affiliations:** 1Division of Gastroenterology, Department of Medicine, School of Medicine, Kurume University, Kurume 830-0011, Japan; 2Department of Orthopedics, School of Medicine, Kurume University, Kurume 830-0011, Japan

**Keywords:** metabolic-associated fatty liver disease, multi-organ diseases, atherosclerotic cardiovascular disease, extrahepatic carcinoma, chronic kidney disease, sarcopenia, extrahepatic diseases, chronic obstructive pulmonary disease, cognitive impairment, thyroid disease

## Abstract

Fatty liver is known to be associated with extra-hepatic diseases including atherosclerotic cardiovascular disease and extra-hepatic cancers, which affect the prognosis and quality of life of the patients. The inter-organ crosstalk is mediated by metabolic abnormalities such as insulin resistance and visceral adiposity. Recently, metabolic dysfunction-associated fatty liver disease (MAFLD) was proposed as a new definition for fatty liver. MAFLD is characterized by the inclusion criteria of metabolic abnormality. Therefore, MAFLD is expected to identify patients at high risk of extra-hepatic complications. In this review, we focus on the relationships between MAFLD and multi-organ diseases. We also describe the pathogenic mechanisms of the inter-organ crosstalk.

## 1. Introduction—What Is MAFLD?

### 1.1. The Relationship between Fatty Liver and Metabolic Abnormality

Fatty liver is classified into nonalcoholic fatty liver disease (NAFLD) and alcoholic fatty liver disease, depending on the amount of alcohol consumed [1]. NAFLD is diagnosed based on imaging and histological findings of fatty liver and the exclusion of secondary causes of fatty liver, such as viral, drug, or alcohol-related diseases. The main causes of NAFLD are binge eating and physical inactivity, and many patients have metabolic abnormalities such as obesity and glucose intolerance. Although these metabolic abnormalities are not required for the diagnosis of NAFLD, metabolic abnormalities are associated with the development of hepatic fibrosis, atherosclerotic cardiovascular disease (ASCVD), and overall prognosis [2,3,4,5].

With the increasing importance of metabolic abnormality in the management of patients with fatty liver, metabolic dysfunction-associated fatty liver disease (MAFLD) has been proposed as a new diagnosis to indicate fatty liver associated with metabolic abnormalities [6]. MAFLD is based on the inclusion criteria for metabolic abnormality and can identify patients at high risk including hepatic fibrosis and ASCVD. Moreover, fatty liver and metabolic abnormality are associated with multi-organ diseases including extra-hepatic carcinoma, chronic obstructive pulmonary disease (COPD), and sarcopenia. This article reviewed the relationships between MAFLD and multi-organ disease.

### 1.2. The Momentum for the Nomenclature of MAFLD and Its Definition

The International Consensus Panel named fatty liver with metabolic abnormalities as MAFLD in May 2020. A joint statement by 32 specialists from 22 countries proposed diagnostic criteria for MAFLD in June 2020 [7], and the guideline was established by the Asian Pacific Association for the Study of the Liver in December 2020 [8]. Furthermore, more than 1000 global multi-stakeholders from more than 134 countries signed a petition in support of the naming and diagnosis of MAFLD in March 2022 [9].

MAFLD is not a simple rebranding of NAFLD but is characterized by the inclusion of metabolic abnormality as a criterion. MAFLD is diagnosed in patients with fatty liver who are: (1) overweight or obese (BMI ≥ 23 in Asia), (2) have type 2 diabetes, or (3) have lean/normal weight (BMI < 23 in Asia) and at least two metabolic abnormalities such as hypertension or dyslipidemia (Figure 1). The metabolic abnormality evaluated in the case of lean and normal weight is based on the diagnostic criteria for metabolic syndrome proposed by The National Cholesterol Education Program (NCEP) Adult Treatment Panel III (ATP III) [10]. MAFLD is expected to identify patients at high risk of developing complications, as metabolic abnormalities are mandatory for its diagnosis.

### 1.3. Differences between MAFLD and NAFLD

The main differences between MAFLD and NAFLD are based on their respective inclusion and exclusion criteria. NAFLD and MAFLD can be distinguished by the presence of metabolic abnormalities and differences in the amount of drinking [11,12]. By changing to MAFLD from NAFLD, patients with moderate or higher alcohol consumption and metabolic abnormality are added (Group 3). On the other hand, patients with low alcohol consumption and no metabolic abnormality are excluded from MAFLD (Group 1) (Figure 2).

### 1.4. The Impact of MAFLD Subtypes on Clinical Outcomes and Treatment

As mentioned in the former section, MAFLD may be divided into three subtypes: diabetic MAFLD (DM-MAFLD), obese MAFLD, and non-obese MAFLD. Although not all the detailed characteristics are known for all of these subtypes, diabetic MAFLD has been well studied. Okada et al. examined the clinical characteristics of different MAFLD subtypes and reported that DM-MAFLD had significantly higher intrahepatic fat mass than other subtypes [13]. In addition, Shao et al. demonstrated that DM-MAFLD is the most predominant subtype associated withmitochondrial dysfunction in serum from 122 MAFLD patients using a metabolomics analysis [14]. Furthermore, Bessho et al. found that DM-MAFLD had a higher insulin resistance than other subtypes, as well as higher levels of high-sensitivity C-reactive protein, and was strongly associated with coronary artery calcification (OR 5.833, *p* < 0.001) [15]. Chen et al. examined the prognosis in 12,377 subjects with the needed information for MAFLD criteria from the U.S. National Health and Nutrition Examination Survey (NHANES III) database and reported that DM-MAFLD was the highest risk subgroup for mortality (HR 1.634, *p* < 0.001) [16]. Therefore, DM-MAFLD has the highest event incidence and is the poorest prognostic subtype of MAFLD, and active care may be recommended.

Although there is still no established pharmacological therapy for NAFLD, recent reports have raised hopes for a therapeutic agent for DM-MAFLD. For instance, Takahashi et al. evaluated the effects of a sodium-glucose cotransporter-2 inhibitor, ipragliflozin, on type 2 diabetes patients with NAFLD, i.e., DM-MAFLD. Liver fibrosis in NAFLD subjects was improved by ipragliflozin with long-term dosing. Despite 33.3% of patients in the control group developing NASH, none of those in the ipragliflozin group developed NASH [17]. In a similar way, semaglutide, a glucagon-like peptide (GLP) 1 receptor agonist, a type of diabetes drug, has also been reported to be effective in improving the clinical appearance and severity of NAFLD, such as hepatic steatosis index and AST to platelet ratio index for DM-MAFLD, in addition to improving glycemic control including homeostatic model assessment for insulin resistance index, fat mass, and visceral adipose tissue [18]. In this regard, the hypothetical effects of combination therapy of SGLT-2 inhibitors and GLP-1 receptor agonists may be a more potent therapeutic agent. Moon et al. mentioned that each of these treatments shows weight loss and protection of the cardiovascular system effect, but the combination is more effective than either of them used separately for improving body weight and glucose tolerance, according to a meta-analysis. Moreover, changes in hepatic matrix delivery including free fatty acids and ketone body metabolism; microbiota changes (mainly GLP-1 receptor agonists); and anti-inflammatory, antifibrotic, and antioxidant effects via adipokine and cytokine were identified [19]. Recently, there has been a lot of interest in incretin receptor agonists, which are GLP-1 receptor agonists and glucose-dependent insulinotropic peptide receptor agonists [20]. These incretin receptor agonists improve multiple metabolic regulations and conditions including insulin tolerance, liver fat, and body weight.

One of the aims of the definition of MAFLD is to find effective treatments by classifying phenotypes of metabolic abnormalities. Based on recent research findings, we may be close to establishing a treatment for MAFLD of the DM type.

## 2. The Relationship with Intrahepatic Lesions

### 2.1. The Relationship with Hepatic Fibrosis

Hepatic fibrosis is a prognostic factor in fatty liver patients [21]. Yamamura et al. compared the ability of MAFLD and NAFLD to enclose hepatic fibrosis and found that MAFLD was the most independent factor relating to hepatic fibrosis (OR 4.401, *p* < 0.0001), identifying approximately 20% more cases of hepatic fibrosis development than NAFLD [22]. Furthermore, MAFLD is associated with more cases of hepatic fibrosis than NAFLD (25.0% vs. 15.5%; *p* = 0.0181), even with low levels of alcohol consumption (OR 4.401, *p* < 0.0001). Similar results have been reported in a prospective large-scale cohort study, the Rotterdam Study, and in a meta-analysis of approximately 10 million individuals [23,24].

### 2.2. The Relationship with Hepatic Carcinogenesis

Myers et al. analyzed the Canton of Geneva Cancer Registry in Switzerland and reported that, of 920 hepatocellular carcinoma (HCC) patients, 8.3% had NAFLD whereas 41% had MAFLD excluding NAFLD, with a particularly marked increase in incidence in women [25]. On the other hand, Vitale et al. investigated a multicenter study in Italy (ITA.LI.CA database) and found that MAFLD accounted for 68.4% of all HCC patients, especially in MAFLD patients without any etiology other than metabolic abnormalities (e.g., viral or alcoholic hepatitis), wherein incidence increased from 3.6% (2002–2003) to 28.9% (2018–2019). HCC in patients with MAFLD and only pure metabolic abnormalities were characterized by larger tumor size and more frequent extrahepatic metastases. However, they reported a lower risk of all-cause and HCC-related mortality in patients with MAFLD-derived HCC compared with patients with non-MAFLD HCC (both *p* < 0.001) [26]. Shimose et al. investigated the effect of MAFLD on advanced HCC in 320 patients treated with lenvatinib in an international, multicenter cohort. They found that overall survival was significantly higher in MAFLD patients than in those without MAFLD (median 21.1 months vs. 15.1 months, *p* = 0.002), and MAFLD was an independent factor for prolonged mortality according to a multi-valuative analysis [27].

## 3. The Relationship between MAFLD and Multi-Organ Disease

### 3.1. The Relationship with Atherosclerotic Cardiovascular Disease

The leading cause of mortality in fatty liver patients is ASCVD [28]. As for the major ASCVD event, it is defined as a recent acute coronary syndrome, history of myocardial infarction, history of ischemic stroke, and symptomatic peripheral arterial disease (history of claudication with ABI < 0.85 or previous revascularization or amputation) as per the 2018 AHA/ACC blood cholesterol guidelines [29]. In addition, cardiovascular disease (CVD) is a generic term for conditions affecting the heart or blood vessels and is a more comprehensive disease concept than ASCVD. CVD includes heart failure, arrhythmias, and congenital heart disease, in addition to diseases included in ASCVD. Therefore, it is important to examine the risk of ASCVD and CVD in MAFLD. Tsutsumi et al. performed a direct view comparison of ASCVD risk between MAFLD and NAFLD using longitudinal health checkup data of 2306 patients [30]. As a result, compared to NAFLD, MAFLD was significantly associated with a higher cardiovascular risk (HR 1.08, 90% CI 1.02–1.15, *p* = 0.014). In addition, they showed that the evidence of metabolic abnormality was more strongly related to ASCVD risk than differences in alcohol consumption in a multivariate analysis. Other studies have reported similar results. For instance, Yoneda et al. found that MAFLD was an independent factor for CVD in a large Japanese insurance coverage cohort database [31]. MAFLD was better associated with CVD than NAFLD, as shown by Lee et al. [32]. Kim et al. also noted that, while NAFLD did not increase overall mortality risk, MAFLD was associated with a higher cardiovascular mortality risk [33]. In addition to extra-hepatic malignancies, ASCVD, and chronic kidney disease (CKD), MAFLD is known to be complicated by a variety of other systemic complications. Quek et al. reviewed 17 articles in a systematic review and meta-analysis (*n* = 12,620,736) and demonstrated that MAFLD increases the risk of stroke (HR 1.55, 95% CI 1.37–1.73), carotid atherosclerosis (OR 1.18, 95% CI 1.00–1.38), peripheral artery disease (OR 1.32, 95% CI 1.05–1.68) as well as obstructive sleep apnea (OR 6.8, 95% CI 1.81–25.6) [34].

Regarding the mechanisms by which MAFLD is highly linked to ASCVD, the roles of individual metabolic abnormalities have been clarified. Drożdż et al. reported that machine learning of MAFLD and ASCVD risk showed that the important clinical variables were hypertension, plaque score, and duration of diabetes [35]. Ye et al. reported that MAFLD is classified into five clusters based on metabolic parameters by cluster analysis with k-means clusters [36]. Patients with severe insulin resistance-related clusters had markedly worse survival and a higher incidence of diabetes and CVD than those in other clusters. In summary, MAFLD is valuable in screening patients at high risk for ASCVD, and it is important to include mandatory metabolic disorders such as diabetes (Table 1).

### 3.2. The Relationship with Extrahepatic Carcinoma

Various investigations have proven that NAFLD is related to a heightened risk of both intra- and extra-hepatic carcinomas, particularly mammary cancers and gastrointestinal cancers [37,38,39,40,41]. Since both fatty liver and metabolic abnormalities such as diabetes have been known to promote carcinogenic effects [37,42,43,44], MAFLD may have a wide range of carcinogenic effects. Liu et al. reported that, in 160,979 MAFLD patients, MAFLD conferred an approximately 7% increased risk of cancer overall among UK Biobank participants [45]. In this study, a higher risk of liver cancer was found in association with MAFLD (HR 1.59, 95% CI 1.28–1.98) as well as an increased risk of extrahepatic cancer (HR 1.07, 95% CI 1.05–1.10) than those with non-MAFLD. These results can be explained by gene variants of PNPLA3, TM6SF2, and MBOAT7 [45]. They also reported the hazard of 24 major types of cancer in 131,449 patients with MAFLD and 221,462 patients without MAFLD in a nationwide cohort analysis [46]. In comparison with non-MAFLD, MAFLD was statistically related to 10 of the 24 cancers evaluated, including uterine (HR 2.36, 95% CI 1.99–2.80), gallbladder (HR 2.20, 95% CI 1.14–4.23), liver (HR 1.81, 95% CI 1.43–2.28), kidney (HR 1.77, 95% CI 1.49–2.11), thyroid (HR 1.69, 95% CI 1.20–2.38), esophagus (HR 1.48, 95% CI 1.25–1.76), pancreas (HR 1.31, 95% CI 1.10–1.56), bladder (HR 1.26, 95% CI 1.11–1.43), breast (HR 1.19, 95% CI 1.11–1.27), and colorectal and anus cancers (HR 1.14, 95% CI 1.06–1.23) [46]. Besides, MAFLD can increase the incidence of colorectal adenomas and cancers [47,48,49] (Table 2).

The pathogenic mechanism of the relationship between fatty liver and cancer risk is that hepatic fat contributes to metabolic disease, insulin resistance, and both systemic and local inflammation, and may promote carcinogenesis [39]. It is therefore reasonable to assume that liver fat is an important contributor not only to liver cancer but also to multi-organ cancers such as gastrointestinal cancers [39]. In addition, the adipose tissue is considered an important regulator of the neoplastic micro-environment for the function of cancer cell growth, metastasis, and recurrence [50].

Adipocytes promote tumorigenesis, including changes in the genetic signature and type of secreted adipokines in the development of mammary carcinoma. Adipocytes also promote the formation of a crown-like structure (CLS) in which adipocytes are surrounded by macrophages. The CLS produces potential mutagens such as reactive oxygen species and reactive oxygen intermediates from dying adipocytes [51]. Cancer-associated adipocytes (CAA), irregular adipocytes, are one of the progressive mediators of breast cancer and their decreased size is a characteristic feature [51]. Dysfunctional adipocytes influence the growth, invasion, and survival of mammary cancers through several mechanisms, including (1) the secretory function of various adipokines, (2) the remodeling of metabolic processes, (3) extracellular matrix remodeling, and (4) the supply of cancer-associated fibroblasts [51]. Therefore, various factors such as MAFLD, CAA, and CLS caused by the adipocytes may promote intra/extrahepatic carcinogenesis (Figure 3).

### 3.3. The Relationship with Chronic Kidney Disease

CKD can be caused by metabolic abnormalities. NAFLD is also an established causal factor for CKD, and this evidence suggests that the risk of CKD is elevated in patients with MAFLD [52,53]. Jung et al. reported, using National Health Insurance Service health examinations in Koria (*n* = 268,946), that MAFLD and NAFLD were both risk categories for CKD (NAFLD: adjusted HR 1.33, 95% CI 1.27–1.39, *p* < 0.001; MAFLD: aHR 1.39, 95% CI 1.33–1.46, *p* < 0.001) [54]. In addition, a higher proportion of participants with MAFLD were identified as being at risk of developing CKD than those with NAFLD. Hashimoto et al. demonstrated that MAFLD was associated with CKD (aOR 1.24, 95% CI 1.14–1.36, *p* < 0.001), whereas patients with fatty liver without metabolic abnormality were not associated with CKD (aHR 1.02, 95% CI 0.79–1.33, *p* = 0.868) [55]. In addition, Tanaka et al. showed that the new onset driver of CKD was MAFLD (aHR 1.12, 95% CI 1.02–1.26, *p* = 0.027) in health examinations over 10 years (*n* = 28,890) [56]. Notably, MAFLD participants had a lower renal function, i.e., the estimated glomerular filtration rate was 74.9 vs. 76.5 mL/min/1.73 m^2^ (*p* < 0.0001), respectively, and a higher prevalence of advanced CKD (stages 3–5: 20.3% vs. 17.8%, *p* = 0.005) than NAFLD participants [57]. Additionally, the risk of prevalent CKD in this population-based cohort was increased almost 1.3-fold by the ultrasound severity of MAFLD, even when adjusted for age, gender, race, use of alcohol, and prediabetes. Metabolic abnormalities, including excess weight, type 2 diabetes mellitus, or metabolic disorders in normal-weight subjects, are related to CKD [55]. These findings led to the conclusion that it is possible to identify high-risk CKD individuals using MAFLD (Table 3).

### 3.4. The Relationship with Sarcopenia

Sarcopenia is defined as muscle wasting and functional impairment [58]. In chronic liver disease, sarcopenia is common and is also an independent prognostic factor [58]. Seo et al. revealed that there was a significant association between MAFLD and sarcopenia compared to non-MAFLD in the Korean cohort study [59] (Table 4). In addition, they performed a sub-analysis to investigate the difference in the risk of sarcopenia among four subgroups of MAFLD: diabetic type MAFLD, lean type MAFLD (≥2 metabolic abnormalities without diabetes), obese type MAFLD (overweight/obese without diabetes, and <2 metabolic disorders), and non-MAFLD. Diabetic MAFLD had the highest risk among these MAFLD sub-groups. One possible reason for this is that patients with diabetes may have various factors associated with muscle protein degradation including insulin resistance and activation of autophagy. In addition, loss of muscle mass and strength may be driven by the impaired loss of motoneurons and an imbalance between denervation and reinnervation in diabetic neuropathy [60]. Thus, the risk for sarcopenia is high in MAFLD patients, in particular, diabetic MAFLD.

Sarcopenia is associated with the outcomes of these fatty liver diseases. Chun et al. examined the relationship between sarcopenia and the risk of liver fibrosis and ASCVD in MAFLD populations [61]. The risk of significant hepatic fibrosis significantly increased in sarcopenic subjects with MAFLD (OR = 4.51 by Fibrosis-4 index and 5.72 by NAFLD fibrosis score) compared with subjects without MAFLD. The risk of ASCVD significantly increased in sarcopenic subjects with MAFLD (OR = 4.08) compared with subjects without MAFLD. Sarcopenia exacerbates hepatic fibrosis in patients with fatty liver by worsening insulin resistance and inflammation [62]. Hepatic fibrosis increases the risk of ASCVD independent of other cardiovascular risk factors [63].

Therefore, sarcopenia exacerbates hepatic fibrosis, and it increases the risk of ASCVD in patients with MAFLD. A possible reason is that sarcopenia is associated with a high level of myostatin. Myostatin is known to produce collagen in hepatic stellate cells, which causes hepatic fibrosis [64]. Myostatin also contributes to the inflammation and remodeling of the vasculature which leads to ASCVD [64].

### 3.5. The Relationship with Thyroid Hormone

It is essential to supply thyroid hormones for healthy growth. By directly influencing the regulation of carbohydrate and lipid metabolism in the liver, thyroid hormones restore the body’s homeostatic condition [65]. Important modulators of lipid metabolism are the thyroid hormone thyroxine and its active derivative triiodothyronine with positive effects on metabolism mainly by activating the thyroid hormone receptor beta (THR-β) isoform in hepatocytes. Possible tissue-specific resistance to circulating thyroid hormone was hypothesized because THR-β mRNA expression was adversely correlated with pathologic NASH severity and further decreased with aging. Therefore, targeting a thyromimetic to the THR-β isoform, which is the predominant liver THR, has the possibility of providing the metabolic advantages of thyroid hormone while preventing undesirable general effects in the cardiac and bone systems, which are mainly controlled by the THR-α isoform [66].

In MAFLD participants, Chen et al. examined the relationship between thyroid function and MAFLD in over ten thousand participants in the NHANES III [67]. They found that hypothyroidism is an independent risk factor for MAFLD (OR = 1.27) and is associated with an increased risk of all-cause (HR = 1.32) and cardiovascular mortality (HR = 1.99) in the MAFLD population (Table 5). In addition, Fan et al. reported that elevated levels of thyroid-stimulating hormones, even in hypothyroid states, are associated with the progression of liver fibrosis [68].

Recently, it has been reported that resmetirom (MGL-3196), a selective THR-β agonist designed to improve non-alcoholic steatohepatitis, resulted in a significant reduction in hepatic fat [69]. Moreover, an analog formula (CS271011) with higher THR-β activity than MGL-3196 has been synthesized and improved dyslipidemia and reduced hepatic steatosis in the diet-induced obesity mouse model [70]. Therefore, THR-β agonists may be a beacon in the treatment of MAFLD.

### 3.6. The Relationship with Other Extrahepatic Diseases

MAFLD has characteristics of systemic inflammation and vascular dysfunction, which are linked to cognitive impairment. Yu et al. examined the relationship between MAFLD and cognitive function based on the database of NHANES III [71]. They found that an increased risk of cognitive impairment was related to MAFLD, particularly in MAFLD groups with significant hepatic stiffness and moderate to severe steatosis. Cognitive decline may be accelerated through chronic inflammation due to adipokine and cytokine imbalances caused by MAFLD. Liver stiffness and steatosis may mediate these roles.

Metabolic abnormalities are related to reflux esophagitis (RE). Fukunaga et al. investigated the influence of MAFLD on the development of RE in a multi-center, longitudinal, observational cohort design involving around 9100 consecutive medical check-ups [72]. They demonstrated that, besides aging hiatal hernia, MAFLD was an independent risk factor for RE. Furthermore, the cumulative incidence of non-obese patients with MAFLD was shown to be significantly higher than that of obese patients with MAFLD by stratified analysis. The most influential metabolic risk factor for the onset of RE was visceral adiposity in non-obese patients with MAFLD. Thus, visceral obesity may increase the risk of RE via changes in both intra-gastric pressure and pro-inflammatory adipocytokine expression.

Liver disease is a common comorbidity in patients with inflammatory bowel disease (IBD). Chen et al. examined the impact of MAFLD on the onset of IBD using the UK Biobank cohort consisting of a total of 221,546 females and 183,867 males [73]. They revealed that patients with MAFLD were associated with a 12% increased risk of IBD compared with those without MAFLD at the baseline. On the other hand, Rodriguez-Duque et al. demonstrated a high prevalence of MAFLD (42.0%) in patients with IBD even though the prevalence of obesity and type 2 diabetes was lower in IBD patients with MAFLD [74], suggesting that a leaky gut may be a causative factor for MAFLD. Thus, these previous studies indicate a bi-directional disease interaction between MAFLD and IBD.

Pulmonary dysfunction and chronic obstructive pulmonary disease (COPD) can be caused by metabolic abnormalities and systemic inflammation. Miao et al. performed a cross-sectional analysis to investigate the relationship between lung function parameters and fibrosis severity in MAFLD [75]. They found that, in health check-up examinees, MAFLD was associated with an impairment of pulmonary function even after adjusting for gender, aging, adiposity measures, smoking status, and significant alcohol intake. Tsutsumi et al. also examined the relationship between MAFLD and COPD using a health check-up database (*n* = 2041) [76]. They found that in addition to the aging process and heavy cigarette smoking, MAFLD was an independent factor for COPD (OR 1.46, 95% CI 1.020–2.101, *p* = 0.0385); they also found that MAFLD was associated with COPD via low-grade inflammation evaluated by the CRP-to-albumin ratio. These findings indicated that MAFLD causes hepatic low-grade inflammation and subsequent COPD.

One disease associated with metabolic syndrome and alcohol consumption is psoriasis, a skin disease. The effect of MAFLD on patients with psoriasis was investigated by Takamura et al. [77]. They demonstrated that MAFLD was observed in 82.6% of patients with psoriasis and was associated with the severity of psoriasis in men. Surprisingly, interleukin (IL)-17 inhibitor, a biological agent for psoriasis, improved the hepatic fibrosis index of psoriasis patients with MAFLD. Thus, IL-17 may be a key molecule linking MAFLD and psoriasis (Table 6).

Osteoporosis has been reported to be of increased risk in patients with NAFLD [78,79]. However, there have been some controversial reports regarding the relationship between BMD and MAFLD, as reported by Li et al. [80]. On the other hand, Liu et al. reported that the risk of osteoporosis is lower in the MAFLD population than in the non-MAFLD population [81]. They discussed that these differences could have been caused by using different diagnostic criteria, which may identify different populations with different bone metabolism, leading to inconsistent results. This means that since MAFLD includes a large number of obese patients, a possible reason may be due to obesity reducing the risk of osteoporosis [82].

These findings suggest that we should be aware of a variety of systemic complications in the clinical practice of patients with MAFLD. In addition, awareness of these relationships is important because liver-targeted treatments may represent a new therapeutic strategy to reduce the risk of a variety of systemic diseases and vice versa.
nutrients-15-01123-t006_Table 6Table 6Major findings on the relationships between MAFLD and extrahepatic organs.OrganMain FindingsReference**Brain**▪MAFLD increased the risk of cognitive impairment (OR 1.47)[71]**Esophagus**▪MAFLD increased the risk of reflux esophagitis (HR 1.21)[72]**Gut**▪MAFLD increased the risk of IBD (HR 1.12)▪The prevalence of MAFLD was high in patients with IBD (OR 1.99–5.55)[73,74]**Lung**▪MAFLD was associated with an impairment of lung function (no description of OR)▪MAFLD was an independent factor in fatty liver patients with COPD (OR 1.46)[75,76]**Skin**▪The prevalence of MAFLD was high in patients with psoriasis (82.6%)▪Hepatic fibrosis was improved by the treatment for psoriasis using IL-17 inhibitor in patients with MAFLD[77]**Bone**▪MAFLD has a lower risk of osteoporosis than those without MAFLD (OR 0.41)[80,81]Color coding of tables is a style used to improve readability. The bold type is used to highlight organ names. Abbreviations: MAFLD, metabolic abnormality-associated fatty liver disease; IBD, inflammatory bowel disease; COPD, chronic obstructive pulmonary disease; IL, interleukin; OR, odds ratio; HR, hazard ratio.


### 3.7. Limitations Regarding the Present MAFLD Research

While we have reviewed the relationships between MAFLD and multiorgan disease, there are some issues to be resolved. These include the still ongoing debate on epidemiology, statistical analysis methods, and study design. These issues need to be discussed in the future.

## 4. Conclusions

MAFLD is a new disease definition that has only been proposed about three years earlier. In summary, however, it has been demonstrated that MAFLD is associated with a wide range of multiple organ diseases, including brain dysfunction, cardiopulmonary disease, and even diseases of the skin and muscles as well as intrahepatic lesions (Figure 4). MAFLD has features of insulin resistance, adipocyte abnormalities, and systemic inflammation. These factors play crucial roles in MAFLD-associated multi-organ crosstalk. Any of these extrahepatic lesions may affect the prognosis and quality of life for patients with MAFLD. In the clinical practice of fatty liver, we should focus on metabolic abnormalities as well as extrahepatic complications. Furthermore, general physicians and hepatologists, who play a primary role in the care of MAFLD patients, need to strengthen their collaboration with other organ specialists.

## Figures and Tables

**Figure 1 nutrients-15-01123-f001:**
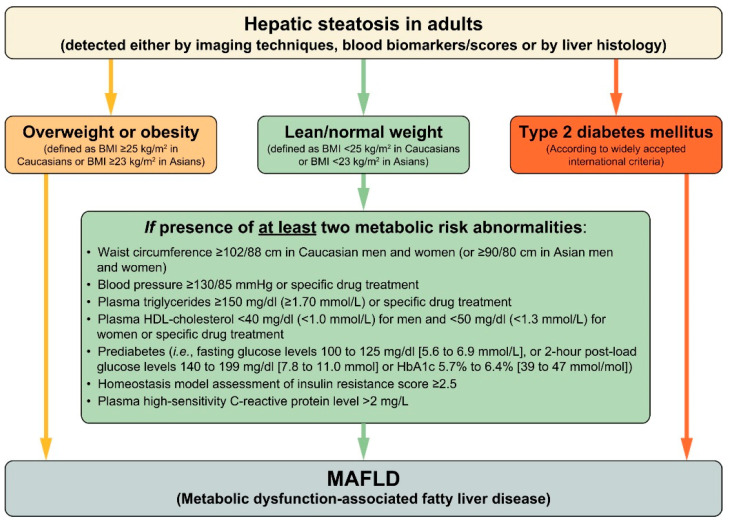
Flowchart of criteria for MAFLD proposed by the global expert consensus panel. Abbreviations: MAFLD, metabolic abnormality-associated fatty liver disease. Adapted with permission from Ref. [7], 2020, Eslam et al.

**Figure 2 nutrients-15-01123-f002:**
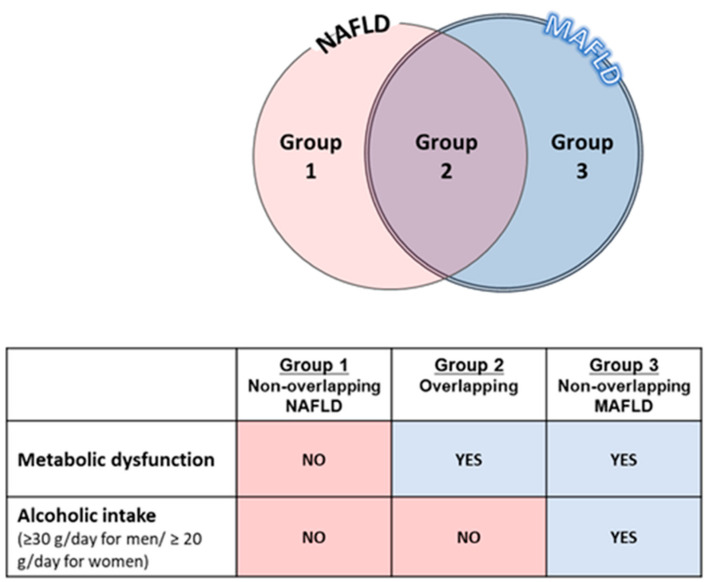
Major differences between MAFLD and NAFLD are based on metabolic abnormalities and alcohol consumption. Abbreviations: MAFLD, metabolic abnormality-associated fatty liver disease. Adapted with permission from Ref. [11], 2022, Kawaguchi et al.

**Figure 3 nutrients-15-01123-f003:**
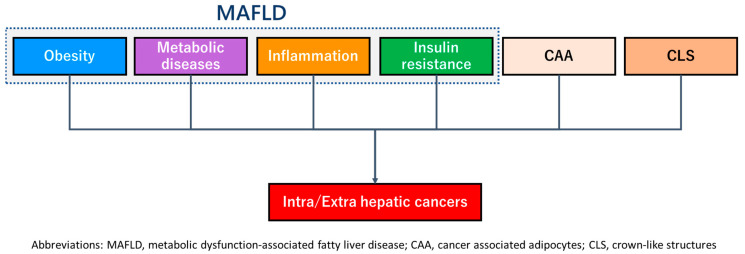
Major pathological mechanisms of multiorgan carcinogenesis. The dotted line indicates features of MAFLD. Abbreviations: MAFLD, metabolic abnormality-associated fatty liver disease; CAA, cancer-associated adipocytes; CLS, crown-like structures.

**Figure 4 nutrients-15-01123-f004:**
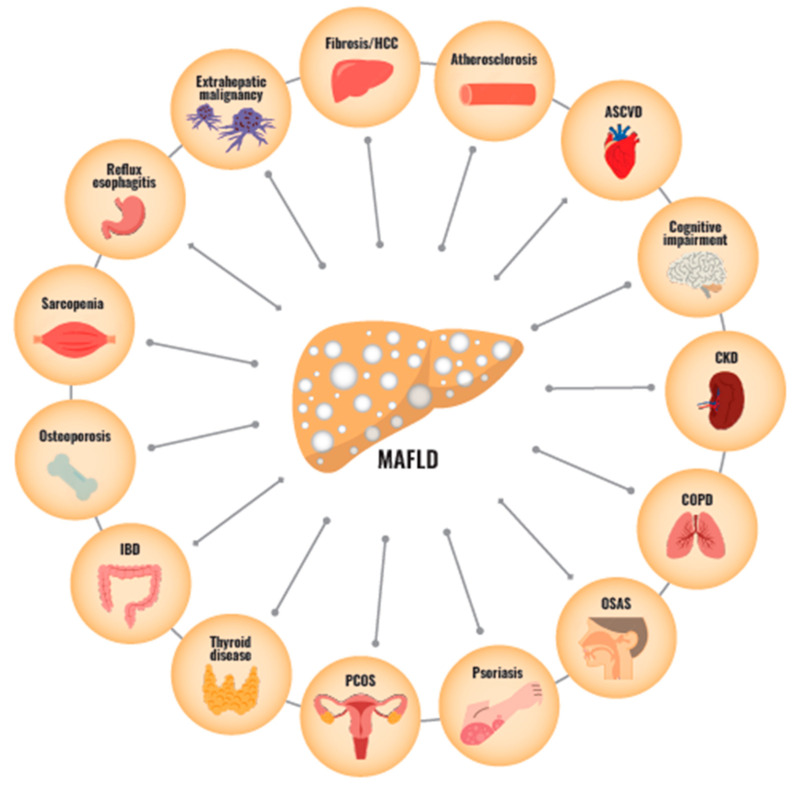
The relationships between MAFLD and multiorgan disease. Abbreviations: MAFLD, metabolic abnormality-associated fatty liver disease; ASCVD, atherosclerotic cardiovascular disease; CKD, chronic kidney disease; COPD, chronic obstructive pulmonary disease; OSAS, obstructive sleep apnea syndrome; PCOS, polycystic ovary syndrome; IBD, inflammatory bowel disease; HCC, hepatocellular carcinoma.

**Table 1 nutrients-15-01123-t001:** Major findings on the relationship between MAFLD and heart.

Organ	Main Findings	Reference
**Heart**	▪ MAFLD increases ASCVD risk more than NAFLD (HR 1.08–1.56)	[30,32]
	▪ MAFLD is an independent driver for CVD (2.69 per 1000 person-years)	[31]
	▪ MAFLD has been correlated with a higher incidence of CVD mortality (HR 1.17)	[33]

Color coding of tables is a style used to improve readability. The bold type is used to highlight organ names. Abbreviations: ASCVD, atherosclerotic cardiovascular disease; MAFLD, metabolic abnormality-associated fatty liver disease; HR, hazard ratio.

**Table 2 nutrients-15-01123-t002:** Major findings on the relationship between MAFLD and extrahepatic cancers.

Organs	Main Findings	Reference
**Extrahepatic cancer**	▪ MAFLD is linked to a higher risk of extrahepatic cancer than non-MAFLD (HR 1.07)	[45]
**Cancer risk in MAFLD versus non-MAFLD**		[46,48]
**Uterus**	HR 2.36	
**Gallbladder**	HR 2.20	
**Kidney**	HR 1.77	
**Thyroid**	HR 1.69	
**Esophagus**	HR 1.48	
**Pancreas**	HR 1.31	
**Bladder**	HR 1.26	
**Breast**	HR 1.26	
**Colorectal and anus**	HR 1.32	

Color coding of tables is a style used to improve readability. The bold type is used to highlight organ names. Abbreviations: MAFLD, metabolic abnormality-associated fatty liver disease; HR, hazard ratio.

**Table 3 nutrients-15-01123-t003:** Major findings on the relationship between MAFLD and the kidney.

Organ	Main Findings	Reference
**Kidney**	▪ MAFLD is at increased risk of CKD (HR 1.12–1.39)	[53,54,55,56,57]

The bold type is used to highlight organ names. Abbreviations: MAFLD, metabolic abnormality-associated fatty liver disease; CKD, chronic kidney disease; HR, hazard ratio.

**Table 4 nutrients-15-01123-t004:** Major findings on the relationship between MAFLD and the muscle.

Organ	Main Findings	Reference
**Muscle**	▪ MAFLD has a high risk of sarcopenia (OR 1.31–1.80)	[59]

The bold type is used to highlight organ names. Abbreviations: MAFLD, metabolic abnormality-associated fatty liver disease; OR, odds ratio.

**Table 5 nutrients-15-01123-t005:** Major findings on the relationship between MAFLD and thyroid function.

Organ	Main Findings	Reference
**Thyroid**	▪ MAFLD is at a high risk of low thyroid function (OR 1.27)	[67]

The bold type is used to highlight organ names. Abbreviations: MAFLD, metabolic abnormality-associated fatty liver disease; OR, odds ratio.

## Data Availability

Not applicable.

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
