# Peer review of "The Inter-Organ Crosstalk Reveals an Inevitable Link between MAFLD and Extrahepatic Diseases"

_nutrients, 2023, doi:10.3390/nu15051123_

Round 1

Reviewer 1 Report

Although the authors of the paper: "The inter-organ crosstalk reveals an inevitable linking MAFLD to extra hepatic diseases"  did an excellent job combining al the information about risk to develop other diseases in patients with MAFLD, minors issues should be addressed in order to publication:

- line 42: COPD was not defined. It is defined in section 2.5 (line 242)

- fig1 had a mistake in "either by"

- section 2 should have an introduction about the principles organs related with MAFLD and their interaction before describing each disease by itself

- CVD: should be described too, although it is part of ASCVD (line 86)

- CKD (line 90) should be remove or section 2.3 should start with CKD instead chronic kidney disease.

- Line 79-81 and 246-248: the authors included their workin this review. These lines should be re-phrase it make it more impersonal rather than saying: " We ......... "

- What's is the meaning of the dotted line in figure 3? Clarify it.

- IBD from line 233 should be moved to line 231 after inflammatory bowel disease.

-Finally, the conclusion is very short and lacking the main aides of this work.

Author Response

To Reviewer 1Thank you very much for your letter regarding our manuscript (nutrients-2226493). We also appreciate your comment, which has helped us to improve our manuscript. In line with your comment, please find below our point-by-point response.

Comments

  1. line 42: COPD was not defined. It is defined in section 2.5 (line 242).

Response: I appreciate your careful peer review. I have corrected the place for the spelling out of COPD. (Line 42, after reflecting the "track changes" functions)

  1. fig1 had a mistake in "either by".

Response: I thank you for your careful peer review. You may be right about your point about this grammar. However, I would like you to understand that this figure is a transcription and cannot be modified by us.

  1. section 2 should have an introduction about the principles of organs related to MAFLD and their interaction before describing each disease by itself.

Response: I appreciate your valuable comment. I have included a new section on MAFLD and intrahepatic lesions. (Line 124-151, after reflecting the "track changes" functions) Again, we appreciate your comment, which has helped us to improve our manuscript.

  1. CVD: should be described too, although it is part of ASCVD (line 86).

Response: I appreciate your pertinent comments. ASCVD is a cardiovascular disease related to atherosclerosis. CVD, on the other hand, is a concept that includes a broader range of cardiovascular diseases than ASCVD, including arrhythmias and heart failure. I have reiterated the cited references and added explanations of the definitions of these terms at the beginning of the section. (Line 155-162, after reflecting the "track changes" functions.)

  1. CKD (line 90) should be remove or section 2.3 should start with CKD instead chronic kidney disease..

Response: I appreciate your careful peer review. I have corrected the spelling out order regarding CKD. (Line 243, after reflecting the "track changes" functions.)

  1. Line 79-81 and 246-248: the authors included their work in this review. These lines should be re-phrase it make it more impersonal rather than saying: " We ......... "

Response: I agree with your comment. It is fair that our description was not in a good form for a review. Thank you for your valuable feedback. (Line 163 and 364, after reflecting the "track changes" functions.)

  1. What's is the meaning of the dotted line in figure 3? Clarify it.

Response: I thank you for your accurate pointing. I added the meaning of the dotted line at the figure legend. (Line 238-239, after reflecting the "track changes" functions.)

Reviewer 2 Report

This was a very well-done review study and the data reported is succinct and well-presented. But still had some scientific issue need further recognized.

1.      In recent study; MAFLD had showed encompasses diverse disease groups with potentially heterogeneous clinical outcomes. Participants with MAFLD were divided into MAFLD-diabetes, MAFLD-overweight/obese, and MAFLD-lean. Different subtype of MAFLD might linkage to different extrahepatic diseases. Authors might consider add this viewpoint in 3.6 section.

2.      In table 3 to Table6, authors need more detailly showed the association of MAFLD with extra-hepatic diseases. Authors need add some data to demonstrate the MAFLD effect in these disease.

I think this paper was still need further revision and add some detail information for future decision. There are major epidemiologic/analytic/study design problems that need to be addressed.

Author Response

To Reviewer 2

Thank you very much for your letter regarding our manuscript (nutrients-2226493). We also appreciate your comment, which has helped us to improve our manuscript. In line with your comment, please find below our point-by-point response.

Comments

  1. In recent study; MAFLD had showed encompasses diverse disease groups with potentially heterogeneous clinical outcomes. Participants with MAFLD were divided into MAFLD-diabetes, MAFLD-overweight/obese, and MAFLD-lean. Different subtype of MAFLD might linkage to different extrahepatic diseases. Authors might consider add this viewpoint in 3.6 section.

Response: I appreciate your spot-on points. Unfortunately, these subtypes are not all known in detail, but diabetic MAFLD has been well studied. Thus, I have cited four papers and mentioned the possibility that DM-MAFLD requires particularly active intervention. I thank you again for your remarks, we were able to improve our article. (Line 80-122, after reflecting the "track changes" functions.)

  1. In table 3 to Table6, authors need more detailly showed the association of MAFLD with extra-hepatic diseases. Authors need add some data to demonstrate the MAFLD effect in these disease.

Response: I agree with your suggestion. We thought it would be better to describe simple information in the tables, but there may not have been enough objective data. (Table 1-6, after reflecting the "track changes" functions.)

I think this paper was still need further revision and add some detail information for future decision. There are major epidemiologic/analytic/study design problems that need to be addressed.

Response: I appreciate your valuable comment. The point you mentioned was lacking in our description, but I think it is very important. We have added them at the end of the main body of the text as limitations of the current study, i.e., issues to be resolved in the future. (Line 397-401, after reflecting the "track changes" functions.)

Round 2

Reviewer 2 Report

Authors had complete response the previous reviewer’s suggestion and revised well. But before accepted this article, authors need improving their clarity of figure. After that, may consider accepted this article.